

# Seed bio-priming with beneficial *Trichoderma harzianum* alleviates cold stress in maize

Mehdi Afrouz[1], R Z. Sayyed[2], Bahman Fazeli-Nasab[3], Ramin Piri[4], Waleed Hassan Almalki[5] and Betty Natalie Fitriatin[6]

[1] Department of Plant Production and Genetic Engineering, University of Mohaghegh Ardabili, Ardabil, Ardabil, Iran
[2] Department of Microbiology, PSGVP Mandal's S I Patil Arts, G B Patel Science and STKV Sangh Commerce College, Shahada, India
[3] Department of Agronomy and Plant Breeding, Agriculture Institute, Research Institute of Zabol, Zabol, Iran
[4] Department of Agronomy and Plant Breeding, Faculty of Agriculture, University of Tehran, Tehran, Iran
[5] Department of Pharmacology, College of Pharmacy, Umm Al-Qura University, Makkah, Saudi Arabia
[6] Department of Soil Science, Padjadjaran University, Jatinangor, Indonesia

Corresponding authors
R Z. Sayyed, sayyedrz@gmail.com
Bahman Fazeli-Nasab, bfazelinasab@gmail.com

## ABSTRACT

Maize is one of the major crops in the world and the most productive member of the *Gramineae* family. Since cold stress affects the germination, growth, and productivity of corn seeds, the present study aimed to investigate the effect of seed biopriming with *Trichoderma harzianum* on the tolerance of two genotypes of maize seedlings to cold stress. This study was conducted in triplicates in factorial experiment with a complete randomized block design (CRBD). The study was conducted in the greenhouse and laboratory of the University of Mohaghegh Ardabili, Ardabil, Iran. Experimental factors include two cultivars (AR68 cold-resistant and KSC703 cold-sensitive maize cultivars), four pretreatment levels (control, biopriming with *T. harzianum*, exogenous *T. harzianum*, and hydropriming), and two levels of cold stress (control and cold at 5 °C) in a hydroponic culture medium. The present study showed that maize leaves' establishment rate and maximum fluorescence (Fm) are affected by triple effects (C*, P*, S). The highest establishment (99.66%) and Fm (994 units) rates were observed in the KP3 control treatment. Moreover, among the pretreatments, the highest (0.476 days) and the lowest (0.182 days) establishment rates were related to P0 and P3 treatments, respectively. Cultivar A showed higher chlorophyll a and b, carotenoid content, and establishment rate compared to cultivar K in both optimal and cold conditions. The highest root dry weight (11.84 units) was obtained in cultivar A with P3 pretreatment. The pretreatments with *T. harzianum* increased physiological parameters and seedling emergence of maize under cold and optimal stress conditions. Pretreatment and cultivar improved catalase activity in roots and leaves. Higher leaf and root catalase activity was observed in the roots and leaves of cultivar K compared to cultivar A. The cold treatment significantly differed in peroxidase activity from the control treatment. Cultivar K showed higher catalase activity than cultivar A. The main effects of pretreatment and cold on polyphenol oxidase activity and proline content showed the highest polyphenol oxidase activity and proline content in hydropriming (H) treatment. Cold treatment also showed higher polyphenol oxidase activity and proline content than cold-free conditions.

## INTRODUCTION

Maize (*Zea mays* L.) is one of the most important cereals in the world (*Zhao et al., 2021*; *Langyan et al., 2021*). Based on the World Food and Agriculture Organization (FAO) reports in 2017, the area under maize cultivation is around 188 million hectares, and the annual production of this plant is about one billion and 60 million tons (*Huzsvai et al., 2020*; *Wani et al., 2021*). The area under cultivation of maize in Iran was about 130 thousand hectares in 2017, with average grain and forage yields of about seven and 45 tons per hectare, respectively (*Bijanzadeh, Naderi & Egan, 2019*; *Zarei et al., 2020*). At the same time, the maize plant is highly efficient in suitable environmental and growth conditions but is very sensitive to environmental stresses such as water shortage and cold (*Sagar et al., 2022*; *Kusale et al., 2021*). Stress at low temperatures limits the production of cereals, especially maize, in cold regions, causing a lot of damage and reducing the growth rate of plants in some years (*Shiyab et al., 2013*).

Crops, mainly maize, are constantly exposed to environmental stresses; that significantly affect crop yields each year. Low temperature is the critical non-biotic stimulus that limits crop production in autumn cultivation in dry and temperate regions (*Sheikh et al., 2022*; *Baba et al., 2021*). Therefore, plant tolerance to cold stress is essential for winter survival, healthy growth, and production. (*Oberschelp et al., 2020*; *Ouellet & Charron, 2013*). Many plant species, mainly tropical and subtropical, including maize, cotton, cucumber, and soybeans, are damaged when exposed to frost at low but above zero temperatures (*Yang, Wu & Cheng, 2011*).

Plant death occurs at low temperatures due to protein deposition, freezing of intercellular water, water movement from the protoplasm to the intercellular space, and the formation of ice crystals within the protoplasm (*Jalilian et al., 2009*). The main effect of cold stress is related to its negative impact on cell membranes, leading to cell dehydration and damage (*Mehrabi & Fazeli-Nasab, 2012*; *Serraj & Sinclair, 2002*). Usually, due to cold stress, young seedlings show signs of reduced leaf development, wilting, and chlorosis. Furthermore, they show accelerated aging and diminished growth in acute cases, finally leading to plant death (*Hussain et al., 2018*). Photosynthesis is affected by low temperatures after a short exposure (between a few h to a few days) so that plant growth and, thus, plant yield are decreased, as fewer carbohydrates will be available to be used in seed production (*Mehrban & Fazeli-Nasab, 2017*; *ORT, 2002*).

Plants have different defense mechanisms against various stresses. Under cold stress conditions, plants accumulate high amounts of compatible solution compounds (*Ahmad, Lim & Kwon, 2013*). These compounds are unchangeable in physiological pH and non-toxic in high concentrations, maintaining osmotic pressure. Moreover, they stabilize the structure of proteins and membranes under stress conditions (*Baba et al., 2021*), which play an essential role in cell adaptation to various stresses. Proline is one of the most important of these compounds, which is synthesized from glutamate and ornithine pathways, and

its synthesis is significantly increased under cold stress conditions (*Ashraf & Foolad, 2007*; *Naidu, Fukai & Gunawardena, 2005*).

Furthermore, the assessment of chlorophyll fluorescence parameters as a suitable and non-destructive method has been used to specify plant species differences in terms of tolerance to environmental stresses and as an essential factor in quantifying the reaction of resistant and cold-sensitive maize cultivars (*Dobrowski et al., 2005*). In this regard, one of the ways to control the effects of this stress is to use biological managing agents such as *Trichoderma* fungi. Stimulation of root growth, production and absorption of nutrients, and thus increasing plant resistance to environmental stresses, are significant features of this fungus (*Hoyos-Carvajal, Orduz & Bissett, 2009*). It contribute for the improvement of the plant resistance towards diseases, the plant's growth as well as its productivity. *Trichoderma* sp. has been used as an effective biocontrol agent against wide range of plant pathogens. It helps in induction of systemic resistance, increase the uptake of nutrient, promote degradation of any pesticides that can bring negative impact to the soil (*Enshasy et al., 2020*).

Inoculation of *T. harzianum* increased fresh weight and lateral roots in tomato plants. *Trichoderma* species are present in all soil types and are the most common cultivable fungi (*Chacón et al., 2007*). *Trichoderma* is used in soil for biological control against soil pathogens, enhancing nutrient uptake, excretion of toxins, promoting sugar and amino acid transfer in plant roots, and induction of plant growth and resistance to environmental stresses (*Zope, Jadhav & Sayyed, 2019*; *Mastouri, Björkman & Harman, 2010*; *Zhang, Gan & Xu, 2019b*). It is reported that using *Trichoderma* strains increases cumin biochemical and morphological parameters under drought stress in greenhouse conditions. The biopriming method treats seeds using a solution containing live microorganisms such as fungi and bacteria. This study used *Trichoderma* fungus for biopriming treatment (*Zhang, Gan & Xu, 2019b*).

*Trichoderma* sp. is among the soil hyphae that are found all over the world. They use a wide range of compounds as sources of carbon and nitrogen. This genus has a global distribution and, as a soil fungi, causes wood rot in some cases (*Chen & Zhuang, 2017*). This study aimed to assess the morphological, physiological, and seedling emergence indicators and their relationship to biological treatments (*Trichoderma* inoculant) in sensitive and cold-resistant maize cultivars and to gain a deep insight into the biotic and abiotic mechanism involved in facilitating the destructive effects of cold stress in plants.

## MATERIALS AND METHODS

This study was carried out in triplicates as a factorial experiment with a CRBD in the research greenhouse and laboratory of the Agriculture and Natural Resources Department, the University of Mohaghegh Ardabili, Ardabil, Iran (38.2106°N, 48.2952°E). Experimental factors included two cultivars (AR68 cold-resistant and KSC703 cold-sensitive maize cultivars), four pretreatment levels (control, biopriming with *Trichoderma*, exogenous *Trichoderma,* and hydropriming (treatment with distilled water), and two levels of cold stress (control and cold at 5 °C) in a hydroponic culture medium.

## Preparation of fungal suspension and exogenous *Trichoderma*-treated seeds

*T. harzianum* was obtained from the culture repository of Department of Plant Production and Genetic Engineering, University of Mohaghegh Ardabili, Ardabil, Iran. The needed suspension population was created in a Petri dish using potato dextrose agar (PDA)) culture media, then cultivated in a zigzag pattern on the PDA culture medium using a loop (laboratory tube) and in an incubator. The spores were placed into Erlenmeyer sterol with culture media after 11 days of fungal growth and profuse sporulation. Next, a fungus suspension was made using a hemocytometer slide (counting spores) with a multiple of $10^7$ in sterile distilled water. Finally, for seed bio-priming, the seeds at room temperature (25−20 °C) for 10 h h in distilled water (control) or fungal suspension (inoculation treatments) were placed (*Fazeli-Nasab et al., 2021*). Following washing with distilled water, the seeds were kept in water for 12 h, then 10 g/kg of seeds *T. harzianum* was added to the seed mass, considering a specified calibrated pH. The seed moisture was increased in the laboratory afterward. For this purpose, first, the seeds were poured into a mesh clot with tiny pores. Then it was placed on a plastic base inside a water-containing sealed container (The volume of water in the container was so much that the water did not come into contact with the mesh cloth containing the seeds). Then the lid of the container was closed so that the moisture inside the container space could be absorbed by the seeds and not let the light shine on them. Finally, for exogenous Trichoderma (suspension) application, 10 g of *T. harzianum* and the nutrient solution were added to the samples. The nutrient solution in this study was prepared using deionized distilled water, according to the instructions provided by *Hoagland & Arnon (1950)*. To irrigate and create natural conditions for the plots, the nutrient solution was prepared, checked for pH before each irrigation, and added to the cultured samples. The nutrient solution pH was adjusted to 6.5 using 1% NaOH and HCl solution. Finally, after pre-testing in the laboratory, collecting data, and analyzing the results, a 48 h cold stress period was applied at the temperature range of 0−5 °C.

Hydropriming treatment was conducted by placing maize seeds in distilled water for 12 h at 25 °C. Next, bio-priming treatment was performed by soaking seeds in water for 12 h and adding 10 g/kg of seeds *T. harzinum* to the seed mass. Then the seed's moisture was increased in the lab by incubating at 25 °C and placing it in a dark condition. After 48 h, they were prepared for a bio-priming test using exogenous *T.* treatment.

## Preparing pots and planting seeds

Preparation of pot and planting of seeds was carried out in triplicates in factorial experiment with a complete randomized block design (CRBD). Following the application of various bio-priming methods in the main experiment, maize seeds were planted in pots containing mineral iron-free medium-size perlite with a grain size of 3 to 5 mm (Kimia-Pars, Tehran, Iran). Perlite was chosen over soil due to its characterisitic features such as light weight, high porosity and high moisture holding capacity compared to soil. Irrigation was carried out using relevant nutrient solutions. Subsequent irrigation with ordinary water was based on the drop of culture medium humidity to 70% FC. The next time Hoogland and Arnon nutrient solution (pH = 6.5) was used again to compensate for nutrient

depletion. Eighteen days after planting, the leaf emergence indicators were measured. At each sampling stage, control and treatment seedlings were sampled separately. The samples were immediately transferred to ice and subsequently refrigerated at −70 °C. The following traits were measured during the experiment under control and stress conditions. The leaf emergence index of the soil plant analysis development (SPAD) was measured with a chlorophyll meter (SPAD-502; Minolta, Chiyoda, Japan). Quantum Performance (Fv (Variable fluorescence)/Fm(maximum fluorescence)) was measured using a fluorometer (American Optics Science, USA). Photosynthetic pigments, including chlorophyll and carotenoids, were also measured (*Lichtenthaler & Wellburn, 1983*). For this purpose, 0.1 g of leaf tissue was gradually crushed with 80% acetone until Chlorophyll entered the acetone solution. Finally, the solution volume reached 2.5 ml by adding 80% acetone. The resulting solution was centrifuged at 400 rpm for 10 min, and then the optical absorption of the supernatant was read at 470, 646.8, and 663.2 nm using a spectrophotometer. The chlorophyll and carotenoids contents were calculated according to the following Eqs. (1)–(4):

$$Chl\ a = 12.25\ A\ 663.2 - 2.798\ A\ 646.8 \tag{1}$$

$$Chlb = 21.50\ A\ 646.8 - 5.10\ A\ 663.2 \tag{2}$$

$$Chl\ Total = Ca + Cb \tag{3}$$

$$Carotenoid\ x = \frac{(1000\ a\ 470 - 1.82Ca - 85.02cb)}{198}. \tag{4}$$

The plants were adapted to darkness in greenhouse conditions using leaf clips for 20 min. Then the fluorescence rate for each treatment was evaluated at a light intensity of 1,000 micromoles (photons) per square meter per second. Fm (maximum fluorescence after illuminating a saturated light pulse to a dark-adapted plant), Fo (the amount of fluorescence after a dark-adapted plant is illuminated by a dim beam of modulated light), Fv (basic or instantaneous fluorescence intensity), and Fv/Fm (maximum quantum yield of photosystem II in the dark-adapted state) was calculated according to the following formula (5):

$$Fv/Fm = \frac{Fm - FO}{Fm}. \tag{5}$$

## Measurement of the physiological traits
### Measurement of soluble protein
The Bradford method measured seedling soluble protein content (*Bradford, 1976*). The extraction buffer was prepared using potassium dihydrogen phosphate ($KH_2PO_4$) and sodium hydroxide (NaOH) based on the Dean method (*Dean, 1985*). For this purpose,

990 µl of Bradford solution was poured into 2 ml microtubes, where 10 µl of the extract was added. After holding for 1 min. to complete the reaction, the solution was poured into 1 cc cuvettes. The absorbance was read at 595 nm using a spectrophotometer.

### Measurement of catalase activity

Catalase activity was measured according to the method of *Cakmak & Horst (1991)*. To calculate the enzyme activity, 100 µl of 30 mM $H_2O_2$ and 100 µl of protein extract were added to 2.8 ml of 25 mM sodium phosphate buffer (pH = 6.8). The absorbance was read for 1 min. using a spectrophotometer at 240 nm.

### Measurement of polyphenol oxidase activity

The polyphenol oxidase activity was measured according to the method of *Raymond, Rakariyatham & Azanza (1993)*. First, some test tubes were placed in a water bath at 40 °C, and then 2.5 ml of 0.2 M phosphate buffer solution with a pH of 6.8 was added to each tube. Then 0.2 ml of 0.02 M pyrogallol was added to the tubes to bring the temperature to 40 °C. In the next step, 2 ml of enzyme extract was added to each tube, and adsorption changes were recorded at 430 nm at 4 min intervals. Enzyme activity was expressed as changes in adsorption rate at 430 nm per mg of protein per min (*Raymond, Rakariyatham & Azanza, 1993*).

### Measurement of proline levels

Proline level was measured using the Bates method (*Bates, Waldren & Teare, 1973*). For this purpose, 0.3 g of fresh tissue plant sample was grounded with 1.5 ml of 3% sulfosalicylic acid and centrifuged. Then 400 µl of the clear supernatant solution was mixed with 2 ml of glacial acetic acid and 2 ml of ninhydrin reagent and placed in a hot water bath (100 °C) for one h. The reaction was stopped in an ice-water bath. Next, 4 ml of toluene was added to this solution and stirred vigorously for about 20 s. Then, the test tubes were kept constant until the two phases were separated. Pink toluene phase absorption was measured at 520 nm with a spectrophotometer. The proline content was calculated using a standard curve.

## Statistical analysis

After conducting the normality test (based on the Shapiro–Wilk test), an analysis of variance (ANOVA) of variables was performed using Statistix 10 software. The mean comparison was performed using the HSD method at a 5% probability level. The graphs were drawn using EXCEL software.

## RESULTS

Analysis of variance of the interactions of the cultivar (V), cold (S), and pretreatment (P) on stand establishment percentage and Fm showed significant results at a 1% probability level (Tables 1 and 2). Moreover, the dual interactions of the cultivar (C) and pretreatment (P) were substantial for root length, root volume, and root dry weight at a 1% probability level and leaf length and leaf dry weight at a 5% probability level. Root volume, root dry weight, chlorophyll content, and Fv/Fm were also significantly affected by the dual interaction of cultivars (C) and cold (S) ($P \leq 0.05$) (Table 1). The double interactions of cold (S) and

**Table 1** Analysis of variance (mean square) for the effect of biological and non-biological inoculation on the measured traits of maize genotypes AR68 and KSC703 under cold conditions.

| Source | DF | Seedling emergence rate | Seedling emergence | Root peroxidase OD change/ mg protein/ min | Root Catalase (mM $H_2O_2$/min) | Root length | Root Volume | Root dry weight |
|---|---|---|---|---|---|---|---|---|
| Stress (S) | 1 | 0.02005 | 5418.75** | 1722.81* | 2.62 | 39.69 | 4.934 | 0.143 |
| Cultivar (C) | 1 | 0.14421** | 1704.08 | 2850** | 1155.03** | 3236.87** | 297.694** | 190.898** |
| Pretreatment (P) | 3 | 0.18591** | 1178.5** | 246.9 | 802.28** | 181.28** | 87.024** | 81.056** |
| S*C | 1 | 0.00002 | 320.33** | 456.31 | 208.35 | 50.37 | 15.221* | 9.915* |
| S*P | 3 | 0.004 | 96.14** | 20.94 | 29.33 | 121.05* | 4.045 | 1.295 |
| C*P | 3 | 0.00091 | 37.81** | 15.36 | 29.31 | 226.22** | 26.354** | 15.471** |
| S*C*P | 3 | 0.00127 | 73.83** | 11.16 | 40.39 | 65.78 | 3.087 | 0.95 |
| Error | 32 | 0.00497 | 1.31 | 298.7 | 67.68 | 40.93 | 2.036 | 2.347 |
| C.V | – | 21.37 | 1.44 | 20.17 | 20.52 | 21.13 | 23.54 | 22.04 |

**Notes.**
[ns], [*] and [**] ns, * and ** are non-significant at 5 and 1% probability levels, respectively.

pretreatment (P) on root length were also statistically significant ($P \leq 0.05$). The main effect of pretreatment (P) was statistically significant on establishment rate, chlorophyll b, carotenoid content, Fv/Fm, Fo, stand establishment percentage, root catalase activity, leaf catalase activity, leaf catalase activity, leaf polyphenol oxidase activity and proline content. Furthermore, the simple effect of cold (S) on chlorophyll b content, root peroxidase activity, leaf polyphenol oxidase activity, proline content, leaf length, and leaf dry weight was statistically significant (Table 2).

## Morphological characteristics
### Establishment percentage and rate
The biopriming with *Tricohderma* positively impacted the growth in maize. Control treatment (Fig. 1A) showed less growth, hydropriming (Fig. 1B) resulted in some improvement while biopriming with *Trichoderma* (Fig. 1C) significantly improved the vigor in maize under green house conditions.

The results on mean comparison showed that C P S triple effects affect the establishment percentage in maize plants. The highest (99.66%) and the lowest (49.66%) establishment percentages were obtained by K-ET-Control treatment and K-C-Cool treatment, respectively (Fig. 2). Among all pretreatments applied, the highest (0.476 days) and the lowest (0.182 days) establishment rates were related to ET and C treatments, respectively. Among the cultivars, cultivar A showed a better establishment rate than cultivar K (Table 3). The highest amount of leaf catalase activity (41.6 mM $H_2O_2$/min) was obtained from the H treatment, which was not statistically significant compared to the B and ET treatments. Cultivar K showed higher leaf catalase activity. Among the five levels of priming, the highest root chlorophyll b (5.31 units) and carotenoid content (142.18 units) were related to ET treatment. Cultivar A significantly differed in chlorophyll b and carotenoid content compared to cultivar K (Table 3).
**Table 2  Analysis of variance (mean square) for the effect of biological and non-biological inoculation on the measured traits of maize genotypes AR68 and KSC703 under cold conditions.**

| Source | DF | Leaf catalase (mM $H_2O_2$/min) | Leaf peroxidase OD change/ mg protein/ min | Leaf polyphenol oxidase OD change/ mg protein/ min | Proline (mg/gFW) | Soluble protein (mg/gFW) | Leaf Length (cm) | Leaf dry weight (g) |
|---|---|---|---|---|---|---|---|---|
| Stress (S) | 1 | 20.216 | 451.043 | 1374.29** | 11.2759** | 0.1788 | 1070.69** | 0.02556* |
| Cultivar (C) | 1 | 378.097* | 969.487 | 98.23 | 1.3897 | 68.1038* | 2965.74** | 0.0462** |
| Pretreatment (P) | 3 | 336.676** | 195.051 | 386.54* | 11.1256** | 23.1507 | 2976.63** | 0.13656** |
| S*C | 1 | 0.791 | 19.802 | 64.96 | 0.3039 | 25.693 | 225.77 | 0.00034 |
| S*P | 3 | 6.487 | 21.655 | 58.46 | 0.4478 | 0.6721 | 118.06 | 0.00481 |
| C*P | 3 | 27.982 | 14.392 | 13.29 | 0.3519 | 0.3327 | 325.81* | 0.01969* |
| S*C*P | 3 | 54.722 | 10.5 | 16.98 | 1.3661 | 0.8094 | 121.59 | 0.00009 |
| Error | 32 | 55.892 | 244.057 | 126.4 | 1.2927 | 14.8712 | 57.26 | 0.00502 |
| C.V | – | 21.46 | 20.07 | 20.20 | 20.35 | 20.10 | 22.66 | 20.99 |

| Source | DF | Chl a (mg/gFW) | Chlb (mg/gFW) | Carotenoeid (mg/gFW) | Fvm | Fo | Fm |
|---|---|---|---|---|---|---|---|
| S | 1 | 78.073 | 5.6708* | 27 | 0.01044 | 3888 | 50830** |
| C | 1 | 645.489** | 11.1814** | 121035** | 0.02623 | 140.1 | 239419** |
| P | 3 | 51.89 | 4.6495** | 10284** | 0.01025* | 28032.7** | 337086** |
| S*C | 1 | 98.734* | 2.6563 | 130 | 0.14719* | 1452 | 419628** |
| S*P | 3 | 6.858 | 0.0995 | 712 | 0.00017 | 1207.8 | 65715** |
| C*P | 3 | 2.649 | 0.1785 | 783 | 0.00126 | 1503.4 | 207105** |
| S*C*P | 3 | 4.635 | 1.5513 | 287 | 0.00023 | 390.4 | 215401** |
| Error | 32 | 19.333 | 0.8511 | 614 | 0.02497 | 974.7 | 19 |
| C.V | – | 20.45 | 20.39 | 22.56 | 20.07 | 20.95 | 0.6 |

**Notes.**
ns, * and ** ns, * and ** are non-significant at 5 and 1% probability levels, respectively.

### Root length

A comparison of the mean interaction of cultivar and pretreatment showed the highest root length (45.75 cm) in cultivar A with ET pretreatment, which was not statistically significant with A-B and A-H. The lowest root length (19.25 cm) was observed in the K cultivar and ET pretreatment (Fig. 3A). Comparing the mean interaction of cold pretreatment, control treatment with ET pretreatment showed the highest root length (37.05 cm). The lowest root length (20.35 cm) was observed in the control treatment (C) (Fig. 3B).

### Root volume

Root volume was also affected by the interaction of cultivar pretreatment and cultivar cold. The highest (12.38 cm$^3$) and the lowest (2.22 cm$^3$) root volumes were observed in cultivar A with ET pretreatment and cultivar K without pretreatment, respectively (Fig. 4A). Based on the comparison of the mean interaction of cultivar cold, it can be shown that cultivar A has a higher root volume than cultivar K, so the highest root volume (87.95 cm$^3$) was observed in cultivar A without cold treatment, showing no significant difference as compared to the

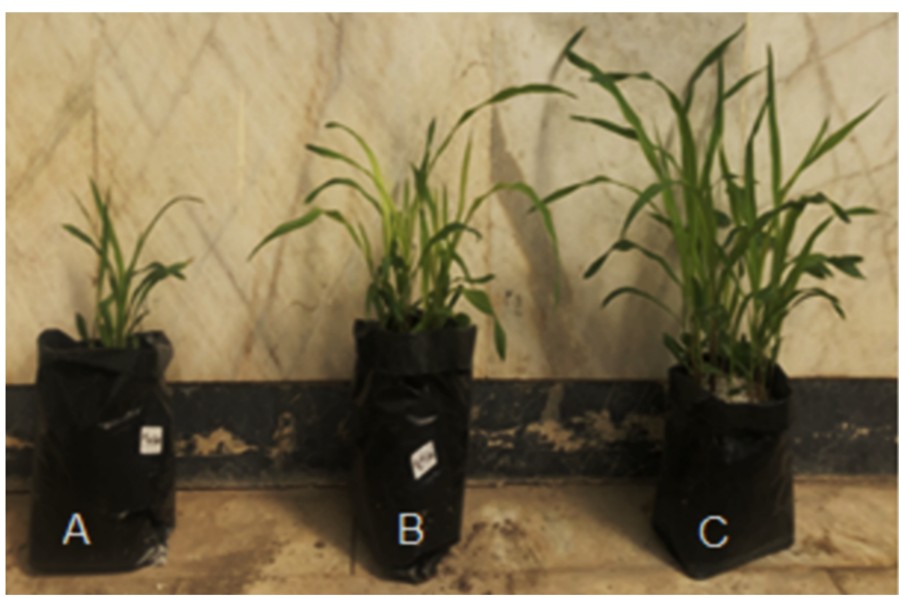

**Figure 1** The maize cultivars in control (A) hydropriming (B) and biopriming with *Trichoderma* (C) under green house conditions.

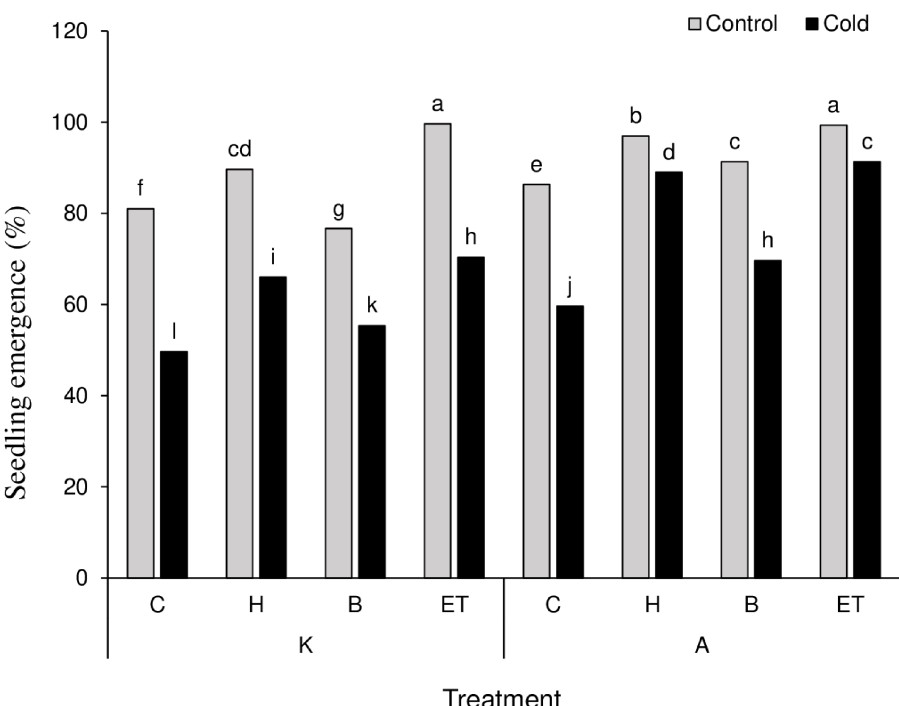

**Figure 2** Comparison of the average of triple interactions (C * P * S) on the percentage of maize plant establishment. Each column and treatment similar letter or letters indicates no significant difference based on the HSD test. C, Control; H, Hydropriming; B, Biopriming and ET, Exogenous *Trichoderma*.

**Table 3  Mean comparison of the effects of pretreatment and seed variety on measured traits of maize.**

| Sources change | FD | Seedling emergence rate (day) | Root catalase (mM H₂O₂/min) | Leaf catalase (mM H₂O₂/min) | Chlb (mg/gFW) | Carotenoeid (mg/gFW) |
|---|---|---|---|---|---|---|
| Pretreatment | C | 0.182$^d$ | 31.23$^c$ | 28.11$^b$ | 3.80$^c$ | 72.19$^c$ |
| | H | 0.370$^b$ | 50.32$^a$ | 41.06$^a$ | 4.59$^{ab}$ | 119.29$^b$ |
| | B | 0.289$^c$ | 36.53$^{bc}$ | 34.94$^a$ | 4.38$^{bc}$ | 105.60$^b$ |
| | ET | 0.476$^a$ | 42.25$^b$ | 35.21$^a$ | 5.31$^a$ | 142.18$^a$ |
| V | A | 0.384$^a$ | 39.85$^b$ | 32.02$^b$ | 5.01$^a$ | 160.03$^a$ |
| | K | 0.275$^b$ | 44.94$^a$ | 37.64$^a$ | 4.04$^b$ | 59.60$^b$ |

**Notes.**
Each column and treatment similar letter or letters is indicative no significant difference based on the HSD test.
C, Control; H, Hydropriming; B, Biopriming; ET, Exogenous *Trichoderma*.

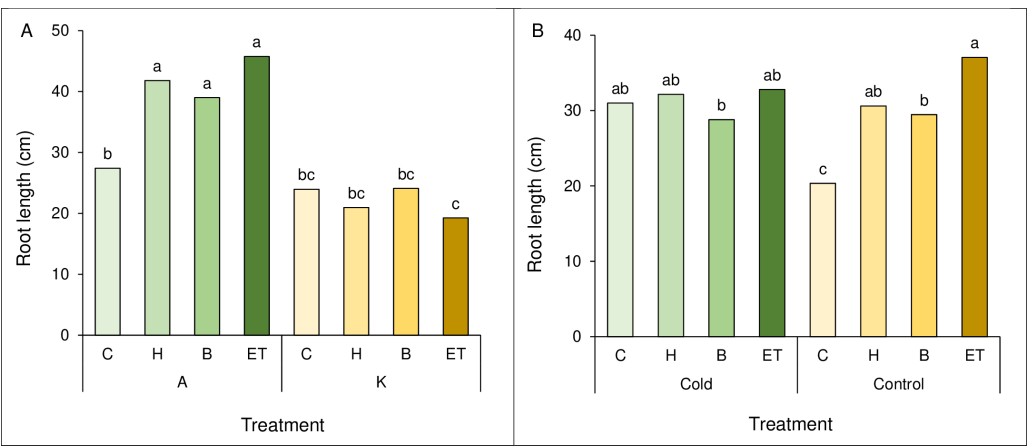

**Figure 3  Comparison of the average interaction (cultivar * pretreatment) (A) and (cold * pretreatment) (B) on maize root length.** Each column and treatment similar letter or letters indicates no significant difference based on the HSD test. C, Control; H, Hydropriming; B, Biopriming and ET, Exogenous *Trichoderma*.

same cultivars with cold treatment. On the other hand, the lowest root volume (26.88 cm³) was observed in the K cultivar in cold-free conditions (Fig. 4B).

## Root dry weight

Comparing the mean interaction of cultivar pretreatment, the highest (11.84 g) and the lowest Root dry weight (3.22 g) was observed in cultivar A with ET pretreatment and cultivar K with no pretreatment, respectively (Fig. 5A). Moreover, a comparison of the mean interaction of the cold * cultivar indicated the best (93.45 g) and the worst (44.47 g) treatments in cultivar A and cultivar K, respectively, in cold-free conditions (Fig. 5B).

## Leaf length and dry weight

A comparison of the mean interaction of cultivar pretreatment on leaf length and leaf dry weight showed that the highest (59 cm) and the lowest leaf length (13.3 cm) in

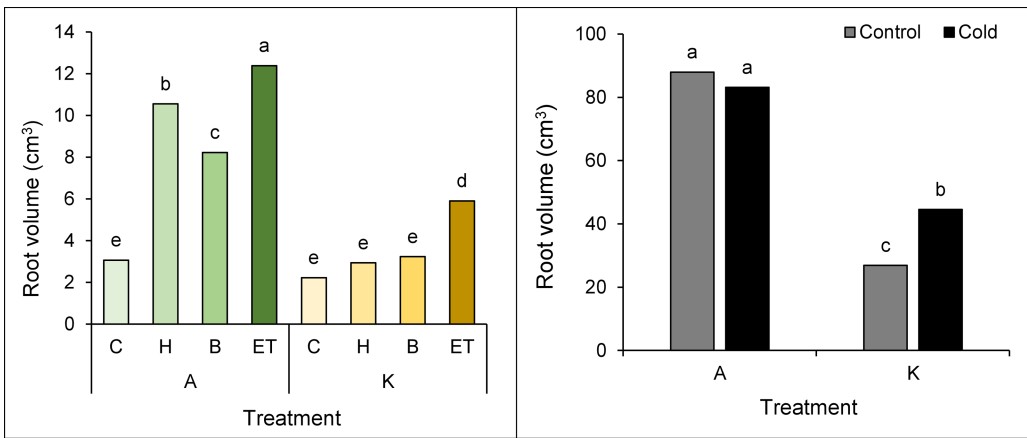

**Figure 4** **Comparison of the average interaction (cultivar* pretreatment) (A) and (cold * pretreatment) (B) on the maize root volume.** Each column and treatment similar letter or letters indicate no significant difference based on the HSD test. C, Control; H, Hydropriming; B, Biopriming and ET, Exogenous *Trichoderma*.

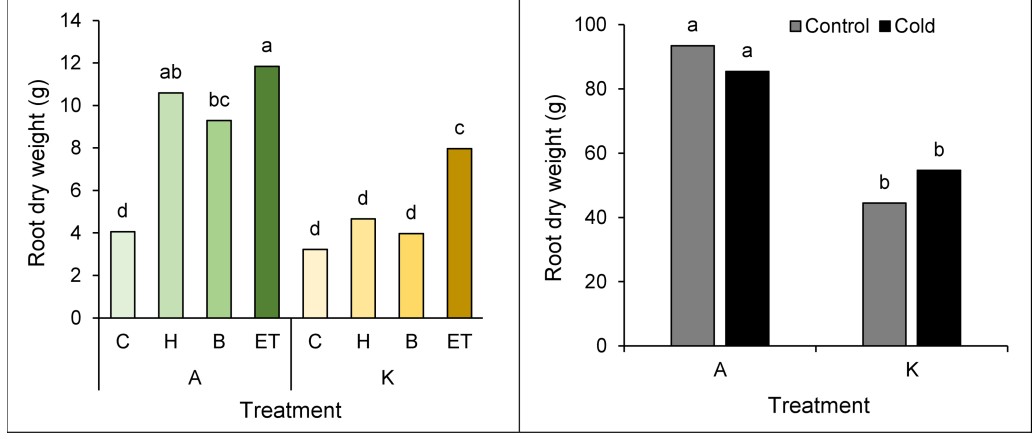

**Figure 5** **Comparison of the average interaction (cultivar * pretreatment) (A) and (cultivar * cold) (B) on the dry weight of maize root.** Each column and treatment similar letter or letters indicates no significant difference based on the HSD test. C, Control; H, Hydropriming; B, Biopriming and ET, Exogenous *Trichoderma*.

cultivar A with H pretreatment and cultivar K with no pretreatment, respectively (Fig. 6A). Furthermore, cultivar K with ET pretreatment and cultivar K with no pretreatment showed the highest (0.463 g) and the lowest (0.134 g) leaf dry weight, respectively (Fig. 6B).

## Physiological characteristics
### Root and leaf catalase activity

A comparison of the mean of the main effects of pretreatment and cultivar on catalase activity in roots and leaves showed the highest root catalase activity (50.32 mM $H_2O_2$/min) and leaf catalase activity (41.6 mM $H_2O_2$/min) was in B treatment among all pretreatments.

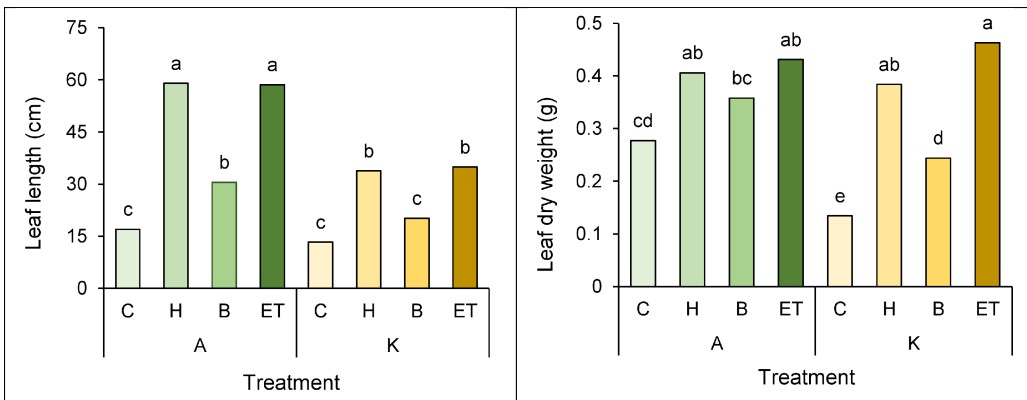

**Figure 6** Comparison of the average interaction (cultivar * pretreatment) on leaf length (A) and leaf dry weight (B) of maize. Each column and treatment similar letter or letters indicates no significant difference based on the HSD test. C, Control; H, Hydropriming; B, Biopriming and ET, Exogenous *Trichoderma*.

Furthermore, more catalase activity was observed in the roots and leaves of cultivar K than in cultivar A (Table 3).

## Root and leaf peroxidase activity

A comparison of the mean of the main effects of cultivar and cold on root peroxidase activity showed significant results. The cold treatment significantly differed from the control treatment based on the peroxidase activity (Fig. 7A). Among cultivars, cultivar K showed higher catalase activity than cultivar A (Fig. 7B). None of the direct, double, and triple interaction effects were significant for leaf peroxidase activity levels (Table 1).

## Leaves polyphenol oxidase activity and proline and soluble proteins content

Comparing the mean of the main effects of pretreatment and cold on polyphenol oxidase activity and proline content showed the highest polyphenol oxidase activity (61.88 μg protein$^{-1}$min$^{-1}$) and proline content (6.49 μg/g of fresh leaf weight) in H treatment. Cold treatment also showed higher polyphenol oxidase activity and proline content than cold-free conditions (Table 4). Moreover, comparing the mean of the main effect of the cultivar on soluble protein content showed higher soluble protein content in cultivar A than in cultivar K (Fig. 8A).

## Vegetation indicators
### Leaves Chlorophyll a, Chlorophyll b, and carotenoids content

A comparison of the mean interaction of cultivar and pretreatment on Chlorophyll a content showed higher Chlorophyll content in both optimal and cold conditions in cultivar A than in cultivar K (Fig. 8B). Our results on the main effects of pretreatment and cultivar on chlorophyll b and carotenoid content indicated the highest chlorophyll b (5.41 units) and carotenoids (142.18 units) content in ET treatment as compared with

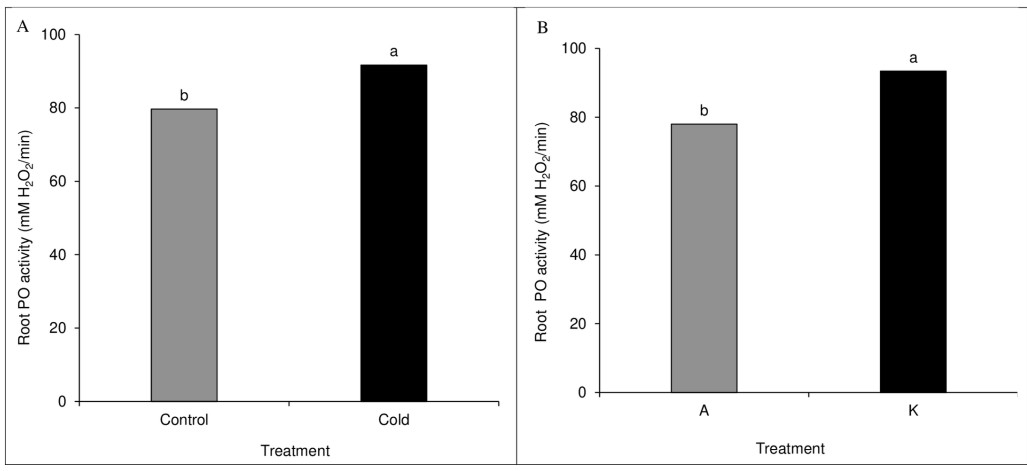

**Figure 7 Comparison of the mean of the main effect of cold (A) and cultivar (B) on maize leaf peroxidase activity.** Each column and treatment similar letter or letters indicates no significant difference based on the HSD test.

**Table 4 Mean comparison of the effects of pretreatment and seed variety on measured traits of maize.**

| Sources change | FD | Polyphenol oxidase ($\mu$g protein$^{-1}$min$^{-1}$) | Proline content ($\mu$g/g of fresh leaf weight) |
|---|---|---|---|
| Pretreatment | C | 49.10[b] | 4.49[b] |
| | H | 61.88[a] | 6.49[a] |
| | B | 53.11[ab] | 5.06[b] |
| | ET | 58.55[a] | 6.28[a] |
| Cold stress | Control | 50.31[b] | 5.10[b] |
| | Cool | 61.01[a] | 6.07[a] |

Notes.
Each column and treatment similar letter or letters is indicative no significant difference based on the HSD test.
C, Control; H, Hydropriming; B, Biopriming; ET, Exogenous *Trichoderma*.

other treatments. Also, cultivar A showed a higher chlorophyll b and carotenoid range than cultivar K (Table 2).

### Fm and Fv/Fm values

Comparison of the mean of cultivar cold interaction on Fv/Fm value in maize leaves showed the highest Fv/Fm value (0.851%) in cultivar K in cold-free conditions, which was not significantly different from that of cultivar A in cold conditions. In cold conditions, the lowest Fv/Fm value (0.693%) was observed in cultivar A (Fig. 8C). Like the establishment percentage, C P S triple effects also affected the Fm value. The highest (994%) and the lowest (143.33%) Fm value was observed in the K-ET-Control treatment and K-C-Cool treatment, respectively (Fig. 8D).

## DISCUSSION

Cold stress is abiotic stress that decreases seedling characteristics and a plant's vegetative growth. Measurement of the establishment percentage as a significant parameter in both

cultivars under stress conditions showed a reduction of cold stress conditions compared to optimal conditions. It is related (*Lahlali et al., 2014*) that cold stress impairs chlorophylls and enzymes activity, protein synthesis, and chloroplast membrane function. Under these conditions, part of plant energy is spent on maintaining the plant in stressful situations instead of on plant growth, reducing its growth. The decrease in germination and establishment percentage is probably due to the interruption at the beginning of the germination process. The interruption seems to be because the seeds need some time to repair stress-induced damage to the membrane and other cell parts, reactivate the antioxidant defense system and prevent oxidative stress. Moreover, repairing such cracks is possible only after water absorption by seeds (*Manisha, Rajak & Jat, 2017*).

*Trichoderma* sp. secrets xylanase and cellulase that can directly stimulate the production of ethylene in the plant for the immune response in the presence of pathogens), antibiotic production, penetration of pathogenic bacteria and fungi, Elimination of poisoning, and increased transfer of sugar and amino acids in plant roots cause induction resistance to stress and biological control of soil diseases (*Harman, 2006*).

In addition, *Trichoderma* sp. exhibits many beneficial features such as solubilization of minerals (*Whipps, 2001*), reduction of phenolic compounds secreted from the roots, acceleration of seed germination, increase in plant resistance under stress conditions (*Lorito et al., 2010*; *Shoresh, Harman & Mastouri, 2010*) and production of the beneficial metabolites like harzianic acid. The increase in growth caused by *T. harzianum* in the plant has been due to the increase in the root area, thus increasing its ability to search for food and, ultimately, access to more food, especially in poor soils (*Yedidia et al., 2001*).

Similar to our results, *Soltani et al. (2009)* found that under stress conditions, the percentage of germination and emergence of resistant seed lots is significantly higher than those of sensitive lots. Our results showed that maize morphological characteristics are affected by cold treatment, and biological pretreatment with *Trichoderma* can improve these morphological indicators, as was previously reported (*Piri et al., 2021*). *Trichoderma* strains promote germination, root growth, plant growth, and establishment percentage due to the production of growth regulators such as auxin, cytokinin, and cytokinin-like molecules such as zeatin gibberellin (*Osiewacz, 2002*). *Alizadeh & Salari (2016)* reported that *Trichoderma* fungus activates the plant's defense system and enhances its resistance to stress by colonizing the plant roots. Also, in a study (*Yedidia et al., 2001*), T203 isolates of *T. harzianum* increased the cucumber shoot length by 45% compared to the control. Possible mechanisms of *Trichoderma*-induced increased shoot length can be through promoting the efficiency of nutrient transport from soil to roots through pathways similar to mycorrhizal fungi (*Jabborova et al., 2022*; *Saboor et al., 2021*; *Najafi et al., 2021*; *Bastami et al., 2021*). Plant peroxidases are significant in plant activities such as interfering with lignin biosynthesis, auxin metabolism, scavenging oxygen free radicals, providing resistance to oxidative stress, wound healing, etc. (*Sobkowiak et al., 2004*). There are strong reasons that cold stress in different plants can result in the accumulation of reactive oxygen species (*Li, Zhang & Shan, 2007*; *Zhang et al., 2019a*). The activity of antioxidant enzymes is not the only plant defense mechanism in reducing oxidative damage; plant biological stimuli through proline synthesis may also reduce the damage caused by free radicals involved

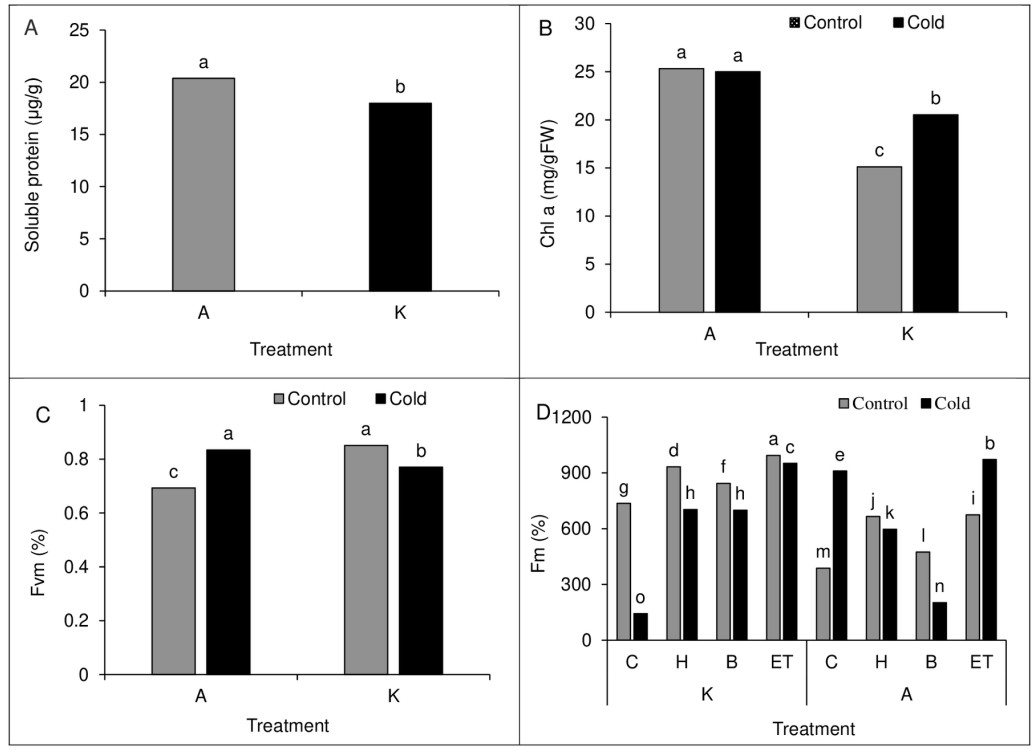

**Figure 8** **(A) Comparison of the mean of the main effect of cultivar on the soluble protein content of maize leaves. (B) Comparison of the average interaction (cultivar * cold) on the chlorophyll a of maize leaves. Each column and treatment similar letter or letters indicates no significant difference based on the HSD test. (C) Comparison of the average interaction (cultivar * cold) on Fvm of maize leaves. Each column and treatment similar letter or letters is indicates no significant difference based on the HSD test. (D) Comparison of the mean of three triple interactions (C * P * S) on the Fm of maize.** Each column and treatment similar letter or letters indicates no significant difference based on the HSD test. C, Control; H, Hydropriming; B, Biopriming and ET, Exogenous *Trichoderma*.

in oxidative stress (*Caverzan et al., 2012*). The effect of inoculation of mustard root with *Trichoderma* under salinity stress has been investigated. In inoculated plants, dry and wet weight, root length and plant height, oil content, amount of micro and macro elements, and activity of catalase, superoxide dismutase, and glutathione peroxidase increased (*Ahmad et al., 2015*). In another research, the influence of inoculation of tomato root with *Trichoderma* has been investigated. The results revealed that the plant inoculated with *Trichoderma* had higher minerals and levels of antioxidant activity than the control plant (*Singh, Singh & Singh, 2013*).

Chlorophyll synthesis is one of the most sensitive processes to temperature changes and is used as a quantitative method to measure cold sensitivity in different plant species (*Colom & Vazzana, 2001*). Chlorophyll stability has been proposed as a criterion for stress resistance in resistant cultivars selection. *Ghassemi-Golezani et al. (2010)* reported a significant decrease in chlorophyll a and b content of seedling leaves with increasing plant stress. *T. harzianum* enhanced cucumber chlorophyll content due to increased cytokinin

production in leaves, producing chloroplasts with expanded granules and chlorophyll (*Lo & Lin, 2002*).

Environmental stresses such as water stress and frost reduce the Fv/Fm ratio to estimate the maximum quantum yield of photosystem II. For example, when plants are exposed to frost stress, leaf metabolism is severely disrupted, and its regenerating ability is reduced, increasing photosystem II vulnerability to light (*Hekneby, Antolín & Sánchez-Díaz, 2006*). Therefore, the degree of cold tolerance in plants can be specified by measuring the Fv/Fm ratio (*Baker & Rosenqvist, 2004*). Similarly, *Cornejo-Ríos et al. (2021)* investigated the effect of tomato seeds pretreatment with *Trichoderma* on traits such as seedling emergence, leaf number, and growth rate, reporting that each of these studied traits was enhanced by seed inoculation due to an increase in chlorophyll content.

Inoculating maize seedlings with *Trichoderma atroviride* increased total chlorophyll, carotenoids, relative water content, superoxide dismutase, ascorbate peroxidase, and catalase activity (*Guler et al., 2016*). The effect of *Trichoderma asperellum* on the cucumber plant has been investigated. The results show that inoculation of cucumber with *Trichoderma* sp. allows the plant to produce more salicylic acid, jasmonic acid, and defense proteins when exposed to stress than non-inoculated plants (*Segarra et al., 2007*). Under drought stress, inoculation of tomatoes with *Trichoderma* improved and reduced the adverse effects of drought stress on the control plant (without inoculation). In plants inoculated with *Trichoderma*, root and shoot growth, chlorophyll, carotenoids, phenols, proline, soluble proteins, auxin, and gibberellin increased (*Mona et al., 2017*). Different species of *Trichoderma*, in addition to the ability to produce auxin with the ability to make organic acids such as gluconic acid, citric acid, and fumaric acid, reduce soil pH and ultimately increase the solubility and absorption of essential micronutrients needed for plant growth such as iron, manganese, magnesium and phosphates (*Vinale et al., 2008*). The effect of growth hormones such as cytokines secreted by *Trichoderma* sp. on plant growth has also been identified (*Benítez et al., 2004*). As the greenhouse study involved statistically designed experiments, the results of green house study can be replicated at field level.

## CONCLUSION

The development and yield of crops are influenced by the genetic characteristics of the plant and its growing conditions. Cold stress can limit the growth of tropical plants like maize by increasing free radicals, producing toxic metabolites, and altering membrane properties. *Trichoderma* bio-treatments improved the growth and emergence of maize under stress conditions by increasing the physiological parameters of these antioxidant enzymes, which can prevent the formation of toxic compounds by modulating or eliminating free radicals. The results of the present study showed that cold stress diminished morphological characteristics and emergence rate in sensitive and cold-resistant maize cultivars by reducing physiological traits and seedling emergence. Nevertheless, the application of biological and non-biological pretreatments was able to moderate the destructive effects of cold stress to some extent. The exogenous application of *Trichoderma* sp. significantly promoted maize's

growth and yield parameters. Application of *Trichoderma* sp. can help to mitigate the damging effects of cold stress while promoting the physiological, morphological and yield parameters of e maize.

### Funding

This work was supported by the Deanship of Scientific Research at Umm Al-Qura University, Makkah, Saudi Arabia, by Grant Code (Project Code: 22UQU4310387DSR17) and the University of Zabol through grant No. IR-UOZ-GR-2735. The funders had no role in study design, data collection and analysis, decision to publish, or preparation of the manuscript.

### Grant Disclosures

The following grant information was disclosed by the authors:
The Deanship of Scientific Research at Umm Al-Qura University, Makkah, Saudi Arabia: 22UQU4310387DSR17.
The University of Zabol:  IR-UOZ-GR-2735.

### Competing Interests

The authors declare that there are no competing interests.

### Author Contributions

- Mehdi Afrouz performed the experiments, analyzed the data, prepared figures and/or tables, and approved the final draft.
- R.Z. Sayyed conceived and designed the experiments, authored or reviewed drafts of the article, and approved the final draft.
- Bahman Fazeli-Nasab conceived and designed the experiments, analyzed the data, prepared figures and/or tables, and approved the final draft.
- Ramin Piri conceived and designed the experiments, authored or reviewed drafts of the article, and approved the final draft.
- Waleed Hassan Almalki analyzed the data, authored or reviewed drafts of the article, fund Aquisition and Revision, and approved the final draft.
- Betty Natalie Fitriatin analyzed the data, prepared figures and/or tables, authored or reviewed drafts of the article, and approved the final draft.

### Data Deposition

   The raw data is available in the Supplementary Files.

### Supplemental Information

Supplemental information for this article can be found online at http://dx.doi.org/10.7717/peerj.15644#supplemental-information.

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
