# Peer review of "Seed bio-priming with beneficial Trichoderma harzianum alleviates cold stress in maize"

_PeerJ, doi:10.7717/peerj.15644_

## Round 0.1 · original submission · Major Revisions

1. Revise the title. After Trichoderma, add the species name, or Trichoderma sp.
2. Add a picture of representative maize plants at the time of harvesting. This will help readers better understand the effects of Trichoderma sp.
3. There is scope for improving the quality of the figures.
4. Add 1-2 statements about the future prospects of this study after the conclusion.
5. Write "maize" instead of "corn".

Reviewer 1 ·

Basic reporting

The paper entitled "Seed bio-priming with beneficial Trichoderma alleviates cold stress in maize" by Mehdi et al. describes the effect of seed inoculation with Trichoderma sp. on maize growth promotion and cold stress mitigation. The authors observed the positive effect of Trichoderma sp. inoculation in growth promotion in maize and in cold stress tolerance.

The English language needs improvments for typos.
All underlying data have been provided; they are robust, statistically sound, & controlled. However, some figures could be improved. See Additional comments area.

Conclusions are well stated, linked to original research question & limited to supporting results.
Results are relevant to the hypotheses. However, units expressions should be revised. See Additional comments area.

Experimental design

no comment.

Validity of the findings

no comment.

Additional comments

Although the work mentioned is of good quality, methodology is scientifically followed, and Results have been discussed in the light of literature, the suggestions given in the following sections may help in the further improvement.

Reviewer Comments
1. The authors have used two terms corn and maize, it is advisable to use the term maize
2. There are few typos and grammar errors
3. Avoid using symbols such as &
4. Expand all the abbreviations at their first appearance
5. Italicize all biological names
6. Hours and other units should be properly mentioned for example hours – h, minutes- min.
7. Objectives should be mentioned in the Abstract
8. The authors have performed measurement of soluble proteins, PPO, catalase, and proline but did not mention these parameters in abstract.
9. Should the genus and species of Trichoderma be mentioned in the abstract?
10. The inoculation method should also be mentioned?
11. How to calculate the optimal concentration of Trichoderma spores?
12. Is Harzianeum correct or harzianum?
13. Mention isp of Trichoderma or write as Trichoderma sp. Throughout the manuscript
14. Line 52-53 - Crops, including corn, ………adverse effects of stresses. Rephrase
15. Line – 54 …temperate regions is one of the essential non-biotic stresses – Rephrase it as
….temperate regions is the most damaging abiotic stress
16. Line 131- Hoagland and Arnon. Mention year
17. Line 136 – What is Hydropriming?
18. Has there been any research on these corn cultivars that are known to be tolerant and sensitive?
19. Line 141 - Preparing pots and planting seeds. Mention experimental design – CRBD?
20. Cakmak's source is not the same as inside the text, it should be corrected.
21. Write all equations equation option from insert tool
22. The serial number of the mushroom should be mentioned.
23. Give GPS location of the study site
24. Why is HSD used instead of LSD?
25. What does FD mean in tables?
26. Chlorophyll a, FVM and FM units should be mentioned in the graphs.
27. Cite more recent references
28. Number of Figures can be minimized. Some of the Figures (Fig 7-10) can be merged as 7a-7d.

·

Basic reporting

The article was write very clear, English redaction is ok, my suggestion is amplified the discussion and conclusions.
Please check the title, Trichoderma sp. or maybe indicate the genera (Trichoderma harzianum).
Introduction:
The corn crops is very important in USA and Europe, but you don't explain what is the importance in your country (Iran right?)....area crops? volume production? market share? economy? money? please support with statistical.
You have clear the importance to T. harzianum, but only have lyne 85-87 to explain it, please amplified. Check the lyne 98 you said Trichoderma inoculants but you used only one Trichoderma harzianum, please clarify it.
Please, check in what else application maybe have the same results, or this is the only raw?

Experimental design

The methodology is clear, the experimental topics is similar to other references, maybe half more description about the inoculate Trichoderma harzianum is a ATCC reference fungi? or you bought it? please indicate the source.
Why used only perlite mineral in the trials? please clarify in the article.
What is the differences between soil and perlite? please indicated in the discussion.
Please explain why used only one gene, maybe a pull the results is more impact.
lyne 127 please cursive letter.

Validity of the findings

This is a trial greenhouse, the explanation of results is ok, please explain if your experimental results is possible to extrapolation to farms or soil with corn corps.
Please check the design table 1, is confused with two title lynes
You have 9 figures!!! please check what is the principal results to show you in the article.
The Trichoderma harzianum is a new gene of Trichoderma? why the question? because only used one reference to specific gene (lyne 572), please, increased the specific references and explain why used only one Trichoderma? maybe is the best? is not clear!!!

Additional comments

Please, check the points.

---

## Round 0.2 · accepted · Accept

The authors have addressed all of the reviewer's comments.

Reviewer 1 ·

Basic reporting

The authors improved the manuscript following the previous commens. I have no further suggestions.

Experimental design

The experimental design is appropriate.

Validity of the findings

The findings are relevant.